# DIVIDE AND CONQUER NETWORKS

**Alex Nowak**
Courant Institute of Mathematical Sciences
Center for Data Science
New York University
New York, NY 10012, USA
alexnowakvila@gmail.com

**David Folqué**
Courant Institute of Mathematical Sciences
Center for Data Science
New York University
New York, NY 10012, USA
david.folque@gmail.com

**Joan Bruna**
Courant Institute of Mathematical Sciences
Center for Data Science
New York University
New York, NY 10012, USA
bruna@cims.nyu.edu

## ABSTRACT

We consider the learning of algorithmic tasks by mere observation of input-output pairs. Rather than studying this as a black-box discrete regression problem with no assumption whatsoever on the input-output mapping, we concentrate on tasks that are amenable to the principle of *divide and conquer*, and study what are its implications in terms of learning.

This principle creates a powerful inductive bias that we leverage with neural architectures that are defined recursively and dynamically, by learning two scale-invariant atomic operations: how to *split* a given input into smaller sets, and how to *merge* two partially solved tasks into a larger partial solution. Our model can be trained in weakly supervised environments, namely by just observing input-output pairs, and in even weaker environments, using a non-differentiable reward signal. Moreover, thanks to the dynamic aspect of our architecture, we can incorporate the computational complexity as a regularization term that can be optimized by backpropagation. We demonstrate the flexibility and efficiency of the Divide-and-Conquer Network on several combinatorial and geometric tasks: convex hull, clustering, knapsack and euclidean TSP. Thanks to the dynamic programming nature of our model, we show significant improvements in terms of generalization error and computational complexity.

## 1 INTRODUCTION

Algorithmic tasks can be described as discrete input-output mappings defined over variable-sized inputs, but this "black-box" vision hides all the fundamental questions that explain how the task can be optimally solved and generalized to arbitrary inputs. Indeed, many tasks have some degree of scale invariance or self-similarity, meaning that there is a mechanism to solve it that is somehow independent of the input size. This principle is the basis of recursive solutions and dynamic programming, and is ubiquitous in most areas of discrete mathematics, from geometry to graph theory. In the case of images and audio signals, invariance principles are also critical for success: CNNs exploit both translation invariance and scale separation with multilayer, localized convolutional operators. In our scenario of discrete algorithmic tasks, we build our model on the principle of *divide and conquer*, which provides us with a form of parameter sharing across scales akin to that of CNNs across space or RNNs across time.

Whereas CNN and RNN models define algorithms with linear complexity, attention mechanisms (Bahdanau et al., 2014) generally correspond to quadratic complexity, with notable exceptions (Andrychowicz & Kurach, 2016). This can result in a mismatch between the intrinsic complexity required to solve a given task and the complexity that is given to the neural network to solve it, which

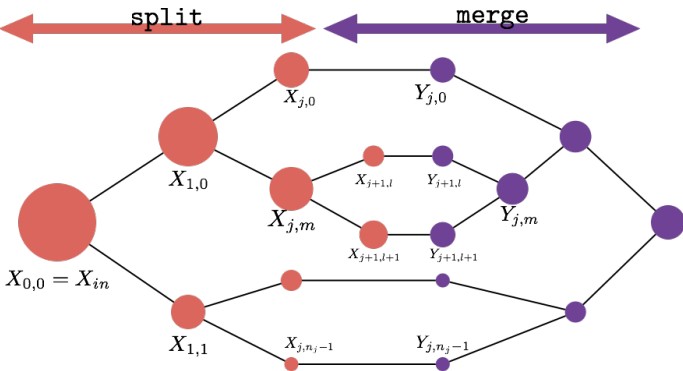

Figure 1: Divide and Conquer Network. The split phase is determined by a dynamic neural network $\mathcal{S}_\theta$ that splits each incoming set into two disjoint sets: $\{X_{j+1,l}, X_{j+1,l+1}\} = \mathcal{S}_\theta(X_{j,m})$, with $X_{j,m} = X_{j+1,l} \sqcup X_{j+1,l+1}$. The merge phase is carried out by another neural network $\mathcal{M}_\phi$ that combines two partial solutions into a solution of the coarser scale: $Y_{j,m} = \mathcal{M}_\phi(Y_{j+1,l}, Y_{j+1,l+1})$; see Section 3 for more details.

may impact its generalization performance. Our motivation is that learning cannot be 'complete' until these complexities match, and we start this quest by first focusing on problems for which the intrinsic complexity is well known and understood.

Our Divide-and-Conquer Networks (DiCoNet ) contain two modules: a *split* phase that is applied recursively and dynamically to the input in a coarse-to-fine way to create a hierarchical partition encoded as a binary tree; and a *merge* phase that traces back that binary tree in a fine-to-coarse way by progressively combining partial solutions; see Figure 1. Each of these phases is parametrized by a single neural network that is applied recursively at each node of the tree, enabling parameter sharing across different scales and leading to good sample complexity and generalisation.

In this paper, we attempt to incorporate the scale-invariance prior with the desiderata to only require weak supervision. In particular, we consider two setups: learning from input-output pairs, and learning from a non-differentiable reward signal. Since our split block is inherently discrete, we resort to policy gradient to train the split parameters, while using standard backpropagation for the merge phase; see Section 5. An important benefit of our framework is that the architecture is dynamically determined, which suggests using the computational complexity as a regularization term. As shown in the experiments, computational complexity is a good proxy for generalisation error in the context of discrete algorithmic tasks. We demonstrate our model on algorithmic and geometric tasks with some degree of scale self-similarity: planar convex-hull, k-means clustering, Knapsack Problem and euclidean TSP. Our numerical results on these tasks reaffirm the fact that whenever the structure of the problem has scale invariance, exploiting it leads to improved generalization and computational complexity over non-recursive approaches.

## 2    RELATED WORK

Using neural networks to solve algorithmic tasks is an active area of current research, but its models can be traced back to context free grammars (Fanty, 1994). In particular, dynamic learning appears in works such as Pollack (1991) and Tabor (2000). The current research in the area is dominated by RNNs (Joulin & Mikolov, 2015; Grefenstette et al., 2015), LSTMs (Hochreiter & Schmidhuber, 1997), sequence-to-sequence neural models (Sutskever et al., 2014; Zaremba & Sutskever, 2014), attention mechanisms (Vinyals et al., 2015b; Andrychowicz & Kurach, 2016) and explicit external memory models (Weston et al., 2014; Sukhbaatar et al., 2015; Graves et al., 2014; Zaremba & Sutskever, 2015). We refer the reader to Joulin & Mikolov (2015) and references therein for a more exhaustive and detailed account of related work.

Amongst these works, we highlight some that are particularly relevant to us. Neural GPU (Kaiser & Sutskever, 2015) defines a neural architecture that acts convolutionally with respect to the input and is applied iteratively $o(n)$ times, where $n$ is the input size. It leads to fixed computational

machines with total $\Theta(n^2)$ complexity. Neural Programmer-Interpreters (Reed & de Freitas, 2015) introduce a compositional model based on a LSTM that can learn generic programs. It is trained with full supervision using execution traces. Directly related, Cai et al. (2017) incorporates recursion into the NPI to enhance its capacity and provide learning certificates in the setup where recursive execution traces are available for supervision. Hierarchical attention mechanisms have been explored in Andrychowicz & Kurach (2016). They improve the complexity of the model from $o(n^2)$ of traditional attention to $o(n \log n)$, similarly as our models. Finally, Pointer Networks (Vinyals et al., 2015b;a) modify classic attention mechanisms to make them amenable to adapt to variable input-dependent outputs, and illustrate the resulting models on geometric algorithmic tasks. They belong to the $\Theta(n^2)$ category class.

## 3 PROBLEM SETUP

### 3.1 SCALE INVARIANT TASKS

We consider tasks consisting in a mapping $\mathcal{T}$ between a variable-sized input set $X = \{x_1, \ldots, x_n\}$, $x_j \in \mathcal{X}$ into an ordered set $Y = \{y_1, \ldots, y_{m(n)}\}$, $y_j \in \mathcal{Y}$. This setup includes problems where the output size $m(n)$ differs from the input size $n$, and also problems where $Y$ is a labeling of input elements. In particular, we will study in detail the case where $Y \subseteq X$ (and in particular $\mathcal{Y} \subseteq \mathcal{X}$).

We are interested in tasks that are self-similar across scales, meaning that if we consider the recursive decomposition of $\mathcal{T}$ as

$$\forall n\, , \forall X\, , |X| = n\, , \quad \mathcal{T}(X) \;=\; \mathcal{M}(\mathcal{T}(\mathcal{S}_1(X)), \ldots, \mathcal{T}(\mathcal{S}_s(X)))\, ,$$
$$|\mathcal{S}_j(X)| \;<\; n\, , \; \cup_{j \leq s} \mathcal{S}_j(X) = X\, , \tag{1}$$

where $\mathcal{S}$ splits the input into smaller sets, and $\mathcal{M}$ merges the solved corresponding sub-problems, then both $\mathcal{M}$ and $\mathcal{S}$ are significantly easier to approximate with data-driven models. In other words, the solution of the task for a certain size $n$ is easier to estimate as a function of the partial solutions $\mathcal{T}(\mathcal{S}_j(X))$ than directly from the input. Under this assumption, the task $\mathcal{T}$ can thus be solved by first splitting the input into $s$ strictly smaller subsets $\mathcal{S}_j(X)$, solving $\mathcal{T}$ on each of these subsets, and then appropriately merging the corresponding outputs together. In order words, $\mathcal{T}$ can be solved by recursion. A particularly simple and illustrative case is the binary setup with $s = 2$ and $\mathcal{S}_1(X) \cap \mathcal{S}_2(X) = \emptyset$, that we will adopt in the following for simplicity.

### 3.2 WEAKLY SUPERVISED RECURSION

Our first goal is to learn how to perform $\mathcal{T}$ for any size $n$, by observing only input-output example pairs $(X^l, Y^l)$, $l = 1 \ldots L$. Throughout this work, we will make the simplifying assumption of binary splitting ($s = 2$), although our framework extends naturally to more general versions. Given an input set $X$ associated with output $Y$, we first define a split phase that breaks $X$ into a disjoint partition tree $\mathcal{P}(X)$:

$$\mathcal{P}(X) = \{X_{j,k}\, ; 0 \leq j < J; 0 \leq k < n_j\}\, , \text{ with } X_{j,k} = X_{j+1,2k} \sqcup X_{j+1,2k+1}\, , \tag{2}$$

and $X = X_{1,0} \sqcup X_{1,1}$. This partition tree is obtained by recursively applying a trainable binary split module $\mathcal{S}_\theta$:

$$\{X_{1,0}, X_{1,1}\} \;=\; \mathcal{S}_\theta(X)\, , \text{ with } X = X_{1,0} \sqcup X_{1,1}\, , \tag{3}$$
$$\{X_{j+1,2k}, X_{j+1,2k+1}\} \;=\; \mathcal{S}_\theta(X_{j,k})\, , \text{ with } X_{j,k} = X_{j+1,2k} \sqcup X_{j+1,2k+1}\, , \; (j < J, k \leq 2^j)\, .$$

Here, $J$ indicates the number of *scales* or depth of recursion that our model applies for a given input $X$, and $\mathcal{S}_\theta$ is a neural network that takes a set as input and produces a binary, disjoint partition as output. Eq. (3) thus defines a hierarchical partitioning of the input that can be visualized as a binary tree; see Figure 1. This binary tree is data-dependent and will therefore vary for each input example, dictated by the current choice of parameters for $\mathcal{S}_\theta$.

The second phase of the model takes as input the binary tree partition $\mathcal{P}(X)$ determined by the split phase and produces an estimate $\hat{Y}$. We traverse upwards the dynamic computation tree determined by the split phase using a second trainable block, the merge module $\mathcal{M}_\phi$:

$$Y_{J,k} \;=\; \tilde{\mathcal{M}}_\phi(X_{J,k})\, , \; (1 \leq k \leq 2^J)\, , \tag{4}$$
$$Y_{j,k} \;=\; \mathcal{M}_\phi(Y_{j+1,2k}, Y_{j+1,2k+1})\, , \; (1 \leq k \leq 2^j, j < J)\, , \text{and } \hat{Y} = \mathcal{M}_\phi(Y_{1,0}, Y_{1,1})\, .$$

Here we have denoted by $\tilde{\mathcal{M}}$ the atomic block that transforms inputs at the leaves of the split tree, and $\mathcal{M}_\phi$ is a neural network that takes as input two (possibly ordered) inputs and merges them into another (possibly ordered) output. In the setup where $Y \subseteq X$, we further impose that $Y_{j,k} \subseteq Y_{j+1,2k} \cup Y_{j+1,2k+1}$ , to guarantee that the computation load does not diverge with $J$.

### 3.3 Learning from non-differentiable Rewards

Another setup we can address with (1) consists in problems where one can assign a cost (or reward) to a given partitioning of an input set. In that case, $Y$ encodes the labels assigned to each input element. We also assume that the reward function has some form of self-similarity, in the sense that one can relate the reward associated to subsets of the input to the total reward.

In that case, (3) is used to map an input $X$ to a partition $\mathcal{P}(X)$, determined by the leaves of the tree $\{X_{J,k}\}_k$, that is evaluated by an external black-box returning a cost $\mathcal{L}(\mathcal{P}(X))$. For instance, one may wish to perform graph coloring satisfying a number of constraints. In that case, the cost function would assign $\mathcal{L}(\mathcal{P}(X)) = 0$ if $\mathcal{P}(X)$ satisfies the constraints, and $\mathcal{L}(\mathcal{P}(X)) = |X|$ otherwise.

In its basic form, since $\mathcal{P}(X)$ belongs to a discrete space of set partitions of size super-exponential in $|X|$ and the cost is non-differentiable, optimizing $\mathcal{L}(\mathcal{P}(X))$ over the partitions of $X$ is in general intractable. However, for tasks with some degree of self-similarity, one can expect that the combinatorial explosion can be avoided. Indeed, if the cost function $\mathcal{L}$ is *subadditive*, i.e.,

$$\mathcal{L}(\mathcal{P}(X)) \leq \mathcal{L}(\mathcal{P}(X_{1,0})) + \mathcal{L}(\mathcal{P}(X_{1,1})) \text{ , with } \mathcal{P}(X) = \mathcal{P}(X_{1,0}) \sqcup \mathcal{P}(X_{1,1}) \text{ ,}$$

then the hierarchical splitting from (3) can be used as an efficient greedy strategy, since the right hand side acts as a surrogate upper bound that depends only on smaller sets. In our case, since the split phase is determined by a single block $\mathcal{S}_\theta$ that is recursively applied, this setup can be cast as a simple fixed-horizon ($J$ steps) Markov Decision Process, that can be trained with standard policy gradient methods; see Section 5.

### 3.4 Computational Complexity as Regularization

Besides the prospect of better generalization, the recursion (1) also enables the notion of computational complexity regularization. Indeed, in tasks that are scale invariant the decomposition in terms of $\mathcal{M}$ and $\mathcal{S}$ is not unique in general. For example, in the sorting task with $n$ input elements, one may select the largest element of the array and query the sorting task on the remaining $n - 1$ elements, but one can also attempt to break the input set into two subsets of similar size using a pivot, and query the sorting on each of the two subsets. Both cases reveal the scale invariance of the problem, but the latter leads to optimal computational complexity ( $\Theta(n \log n)$ ) whereas the former does not ($\Theta(n^2)$). Therefore, in a trainable divide-and-conquer architecture, one can regularize the split operation to minimize computational complexity; see Appendix A.

## 4 Neural Models for $\mathcal{S}$ and $\mathcal{M}$

### 4.1 Split

The split block $\mathcal{S}_\theta$ receives as input a variable-sized set $X = (x_1, \ldots, x_n)$ and produces a binary partition $X = X_0 \sqcup X_1$. We encode such partition with binary labels $z_1 \ldots z_n, z_m \in \{0, 1\}, m \leq n$. These labels are sampled from probabilities $p_\theta(z_m = 1 \mid X)$ that we now describe how to parametrize. Since the model is defined over sets, we use an architecture that certifies that $p_\theta(z_m = 1 \mid X)$ are invariant by permutation of the input elements. The *Set2set* model (Vinyals et al., 2015a) constructs a nonlinear set representation by cascading $R$ layers of

$$h_m^{(1)} = \rho\left(B_{1,0}x_m + B_{2,0}\mu(X)\right) \text{ , } h_m^{(r+1)} = \rho\left(B_{1,r}h_m^{(r)} + n^{-1}B_{2,r}\sum_{m' \leq n}h_{m'}^{(r)}\right) \text{ ,} \qquad (5)$$

with $m \leq n$, $r \leq R$, $h_m^{(r)} \in \mathbb{R}^d$, and $p_\theta(z_m = 1 \mid X) = \text{Sigm}(b^T h_m^{(R)})$. The parameters of $\mathcal{S}_\theta$ are thus $\theta = \{B_0, B_{1,r}, B_{2,r}, b\}$. In order to avoid covariate shifts given by varying input set distributions and sizes, we consider a normalization of the input that standardizes the input variables $x_j$ and feeds the mean and variance $\mu(X) = (\mu_0, \sigma)$ to the first layer. If the input has some structure,

for instance $X$ is the set of vertices of a graph, a simple generalization of the above model is to estimate a graph structure specific to each layer:

$$h_m^{(1)} = \rho\left(B_{1,0}x_m + B_{2,0}\mu(X)\right) , \ h_m^{(r+1)} = \rho\left(B_{1,r}h_m^{(r)} + n^{-1}B_{2,r}\sum_{m'\leq n}A_{m,m'}^{(r)}h_{m'}^{(r)}\right) , \quad (6)$$

where $A^{(r)}$ is a similarity kernel computed as a symmetric function of current hidden variables: $A_{m,m'}^{(r)} = \varphi(h_m^{(r)}, h_{m'}^{(r)})$. This corresponds to the so-called graph neural networks (Scarselli et al., 2009; Duvenaud et al., 2015) or neural message passing Gilmer et al. (2017).

Finally, the binary partition tree $\mathcal{P}(X)$ is constructed recursively by first computing $p_\theta(z \mid X)$, then sampling from the corresponding distributions to obtain $X = X_0 \sqcup X_1$, and then applying $\mathcal{S}_\theta$ recursively on $X_0$ and $X_1$ until the partition tree leaves have size smaller than a predetermined constant, or the number of scales reaches a maximum value $J$. We denote the resulting distribution over tree partitions by $\mathcal{P}(X) \sim \mathbf{S}_\theta(X)$.

## 4.2 MERGE

### 4.2.1 MERGE MODULE

The merge block $\mathcal{M}_\phi$ takes as input a pair of sequences $Y_0, Y_1$ and produces an output sequence $O$. Motivated by our applications, we describe first an architecture for this module in the setup where the output sequence is indexed by elements from the input sequences, although our framework can be extended to more general setups seamlessly. in Section C.

Given an input sequence $Y$, the merge module computes a stochastic matrix $\Gamma_Y$ (where each row is a probability distribution) such that the output $O$ is expressed by binarizing its entries and multiplying it by the input:

$$O = \mathcal{M}_\phi(Y_0, Y_1) = \bar{\Gamma}\left(\begin{array}{c} Y_0 \\ Y_1 \end{array}\right) , \text{ with } \bar{\Gamma}_{s,i} = \left\{\begin{array}{ll} 1 & \text{if } i = \arg\max_{i'} p_s(i') . \\ 0 & \text{otherwise.} \end{array}\right. \quad (7)$$

Since we are interested in weakly supervised tasks, the target output only exists at the coarsest scale of the partition tree. We thus also consider a generative version $\mathcal{M}_\phi^g$ of the merge block that uses its own predictions in order to sample an output sequence. The initial merge operation at the finest scale $\tilde{\mathcal{M}}$ is defined as the previous merge module applied to the input $(X_{J,k}, \emptyset)$. This merge module operation can be instantiated with Pointer Networks (Vinyals et al., 2015b) and with Graph Neural Networks/ Neural Message Passing (Gilmer et al., 2017; Kearnes et al., 2016; Bronstein et al., 2016).

**Pointer Networks** We consider a Pointer Network (PtrNet) (Vinyals et al., 2015b) to our input-output interface as our merge block $\mathcal{M}_\phi$. A PtrNet is an auto-regressive model for tasks where the output sequence is a permutation of a subsequence of the input. The model encodes each input sequence $Y_q = (x_{1,q}, \ldots, x_{n_q,q})$, $q = 0, 1$, into a global representation $e_q := e_{q,n_q}$, $q = 0, 1$, by sequentially computing $e_{1,q}, \ldots, e_{n_q,q}$ with an RNN. Then, another RNN decodes the output sequence with initial state $d_0 = \rho(A_0 e_0 + A_1 e_1)$, as described in detail in Appendix D. The trainable parameters $\phi$ regroup to the RNN encoder and decoder parameters.

**Graph Neural Networks** Another use case of our Divide and Conquer Networks are problems formulated as paths on a graph, such as convex hulls or the travelling salesman problem. A path on a graph of $n$ nodes can be seen as a binary signal over the $n \times n$ edge matrix. Leveraging recent incarnations of Graph Neural Networks/ Neural Message Passing that consider both node and edge hidden features (Gilmer et al., 2017; Kearnes et al., 2016; Bronstein et al., 2016), the merge module can be instantiated with a GNN mapping edge-based features from a bipartite graph representing two partial solutions $Y_0, Y_1$, to the edge features encoding the merged solution. Specifically, we consider the following update equations:

$$V_m^{(k+1)} = \rho\left(\beta_1 V_m^{(k)} + \sum_{m'} A_{m,m'}^{(k)} V_{m'}^{(k)}\right) \quad (8)$$

$$A_{m,m'}^{(k+1)} = \varphi(V_m^{(k+1)}, V_{m'}^{(k+1)}) , \quad (9)$$

where $\varphi$ is a symmetric, non-negative function parametrized with a neural network.

### 4.2.2 RECURSIVE MERGE OVER PARTITION TREE

Given a partition tree $\mathcal{P}(X) = \{X_{j,k}\}_{j,k}$, we perform a merge operation at each node $(j,k)$. The merge operation traverses the tree in a fine-to-coarse fashion. At the leaves of the tree, the sets $X_{J,k}$ are transformed into $Y_{J,k}$ as $Y_{J,k} = \mathcal{M}_\phi^g(X_{J,k}, \emptyset)$, and, while $j > 0$, these outputs are recursively transformed along the binary tree as $Y_{j,k} = \mathcal{M}_\phi^g(Y_{j+1,2k}, Y_{j+1,2k+1})$, $0 < j < J$, using the auto-regressive version, until we reach the scale with available targets: $\hat{Y} = \mathcal{M}_\phi(Y_{1,0}, Y_{1,1})$. At test-time, without ground-truth outputs, we replace the last $\mathcal{M}_\phi$ by its generative version $\mathcal{M}_\phi^g$.

### 4.2.3 BOOTSTRAPPING THE MERGE PARTITION TREE

The recursive merge defined at (4.2.2) can be viewed as a factorized attention mechanism over the input partition. Indeed, the merge module outputs (21) include the stochastic matrix $\Gamma = (p_1, \ldots, p_S)$ whose rows are the $p_s$ probability distributions over the indexes. The number of rows of this matrix is the length of the output sequence and the number of columns is the length of the input sequence. Since the merge blocks are cascaded by connecting each others outputs as inputs to the next block, given a hierarchical partition of the input $\mathcal{P}(X)$, the overall mapping can be written as

$$\hat{Y} = \left(\prod_{j=0}^{J} \widetilde{\Gamma}_j\right) \begin{bmatrix} Y_{J,0} \\ \vdots \\ Y_{J,n_J} \end{bmatrix}, \text{ with } \widetilde{\Gamma}_0 = \bar{\Gamma}_{0,0}, \widetilde{\Gamma}_1 = \begin{pmatrix} \bar{\Gamma}_{1,0} & 0 \\ 0 & \bar{\Gamma}_{1,1} \end{pmatrix}, \widetilde{\Gamma}_J = \begin{pmatrix} \bar{\Gamma}_{J,0} & 0 & \cdots \\ 0 & \bar{\Gamma}_{J,1} & \ddots \\ 0 & \cdots & \bar{\Gamma}_{J,n_J} \end{pmatrix}.$$
(10)

It follows that the recursive merge over the binary tree is a specific reparametrization of the global permutation matrix, in which the permutation matrix has been decomposed into a product of permutations dictated by the binary tree, indicating our belief that many routing decisions are done locally within the original set. The model is trained with maximum likelihood using the product of the non-binarized stochastic matrices. Lastly, in order to avoid singularities we need to enforce that $\log p_{s,t_s}$ is well-defined and therefore that $p_{s,t_s} > 0$. We thus regularize the quantization step (21) by replacing $0, 1$ with $\epsilon^{1/J}, 1 - n\epsilon^{1/J}$ respectively. We also found useful to binarize the stochastic matrices at fine scales when the model is close to convergence, so gradients are only sent at coarsest scale. For simplicity, we use the notation $p_\phi(Y \mid \mathcal{P}(X)) = \prod_{j=0}^{J} \widetilde{\Gamma}_j = \mathbf{M}_\phi(\mathcal{P}(X))$, where now the matrices $\tilde{\Gamma}_j$ are not binarized.

## 5 TRAINING

This section describes how the model parameters $\{\theta, \phi\}$ are estimated under two different learning paradigms. Given a training set of pairs $\{(X^l, Y^l)\}_{l \leq L}$, we consider the loss

$$\mathcal{L}(\theta, \phi) = \frac{1}{L} \sum_{l \leq L} \mathbb{E}_{\mathcal{P}(X) \sim \mathbf{S}_\theta(X)} \log p_\phi(\hat{Y} = Y^l \mid \mathcal{P}(X^l)), \text{ with } p_\phi(Y \mid \mathcal{P}(X)) = \mathbf{M}_\phi(\mathcal{P}(X)).$$
(11)

Section 4.2 explained how the merge phase $\mathbf{M}_\phi$ is akin to a structured attention mechanism. Equations (10) show that, thanks to the parameter sharing and despite the quantizations affecting the finest leaves of the tree, the gradient

$$\nabla_\phi \log p_{\theta,\phi}(Y \mid X) = \mathbb{E}_{\mathcal{P}(X) \sim \mathbf{S}_\theta(X)} \nabla_\phi \log \mathbf{M}_\phi(\mathcal{P}(X))$$
(12)

is well-defined and non-zero almost everywhere. However, since the split parameters are separated from the targets through a series of discrete sampling steps, the same is not true for $\nabla_\theta \log p_{\theta,\phi}(Y \mid X)$. We therefore resort to the identity used extensively in policy gradient methods. For arbitrary $F$ defined over partitions $\mathcal{X}$, and denoting by $f_\theta(\mathcal{X})$ the probability density of the random partition $\mathbf{S}_\theta(X)$, we have

$$\nabla_\theta \mathbb{E}_{\mathcal{P}(X) \sim \mathbf{S}_\theta(X)} F(\mathcal{P}(X)) = \sum_{\mathcal{X}} F(\mathcal{X}) \nabla_\theta f_\theta(\mathcal{X}) = \sum_{\mathcal{X}} F(\mathcal{X}) f_\theta(\mathcal{X}) \nabla_\theta \log f_\theta(\mathcal{X}) = \mathbb{E}_{\mathcal{X} \sim \mathbf{S}_\theta(X)} F(\mathcal{X}) \nabla_\theta \log f_\theta(\mathcal{X})$$

$$\approx \frac{1}{S} \sum_{\tilde{X}^{(s)} \sim \mathbf{S}_\theta(X)} F(\mathcal{X}^{(s)}) \nabla_\theta \log f_\theta(\mathcal{X}^{(s)}).$$
(13)

Since the split variables at each node of the tree are conditionally independent given its parent, we can compute $\log f_\theta(\mathcal{P}(X))$ as

$$\log f_\theta(\mathcal{P}(X)) = \sum_{j=1}^{J} \sum_{k \leq n_j} \sum_{m \leq |X_{j,k}|} \log p_\theta(z_{m,j,k} \mid X_{j-1,k/2}) \, .$$

By plugging $F(\mathcal{P}(X)) = \log p_\phi(Y \mid \mathcal{P}(X))$ we thus obtain an efficient estimation of $\nabla_\theta \mathbb{E}_{\mathcal{P}(X) \sim \mathbf{S}_\theta(X)} \log p_\phi(Y \mid \mathcal{P}(X))$.

From (13), it is straightforward to train our model in a regime where a given partition $\mathcal{P}(X)$ of an input set is evaluated by a black-box system producing a reward $R(\mathcal{P}(X))$. Indeed, in that case, the loss becomes

$$\mathcal{L}(\theta) = \frac{-1}{L} \sum_{l \leq L} \mathbb{E}_{\mathcal{P}(X) \sim \mathbf{S}_\theta(X)} R(\mathcal{P}(X)) \, , \tag{14}$$

which can be minimized using (13) with $F(\mathcal{P}(X)) = R(\mathcal{P}(X))$.

## 6 EXPERIMENTS

We present experiments on three representative algorithmic tasks: convex hull, clustering and knapsack. We also report additional experiments on the Travelling Salesman Problem in the Appendix. The hyperparameters used for each experiment can be found at the Appendix. [1]

### 6.1 CONVEX HULL

The convex hull of a set of $n$ points $X = \{x_1, \ldots, x_n\}$ is defined as the extremal set of points of the convex polytope with minimum area that contains them all. The planar (2d) convex hull is a well known task in discrete geometry and the optimal algorithm complexity is achieved using divide and conquer strategies by exploiting the self-similarity of the problem. The strategy for this task consists of splitting the set of points into two disjoint subsets and solving the problem recursively for each. If the partition is balanced enough, the overall complexity of the algorithm amounts to $\Theta(n \log n)$. The split phase usually takes $\Theta(n \log n)$ because each node involves a median computation to make the balanced partition property hold. The merge phase can be done in linear time on the total number of points of the two recursive solutions, which scales logarithmically with the total number of points when sampled uniformly inside a polytope (Dwyer, 1988).

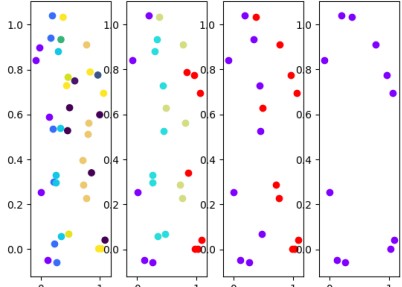 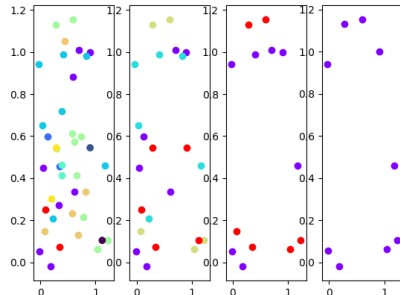

Figure 2: DiCoNet outputs at test time for $n = 50$. The colors indicate the partitions at a given scale. Scales go fine-to-coarse from left to right. *Left:* Split has already converged using the rewards coming from the merge. It gives disjoint partitions to ease the merge work. *Right:* DiCoNet with random split phase. Although the performance degrades due to the non-optimal split strategy, the model is able to output the correct convex hull for most of the cases.

---

[1] Publicly available code to reproduce all results are at `https://github.com/alexnowakvila/DiCoNet`

We test the DiCoNet on the setting consisting of $n$ points sampled in the unit square $[0,1]^2 \subset \mathbb{R}^2$. This is the same setup as Vinyals et al. (2015b). The training dataset has size sampled uniformly from 6 to 50. The training procedure is the following; we first train the baseline pointer network until convergence. Then, we initialize the DiCoNet merge parameters with the baseline and train both split and merge blocks. We use this procedure in order to save computational time for the experiments, however, we observe convergence even when the DiCoNet parameters are initialized from scratch. We supervise the merge block with the product of the continuous $\Gamma$ matrices. For simplicity, instead of defining the depth of the tree dynamically depending on the average size of the partition, we fix it to 0 for 6-12, 1 for 12-25 and 2 for 25-50; see Figure 2 and Table 1.

|                       | n=25     | n=50     | n=100    | n=200    |
|-----------------------|----------|----------|----------|----------|
| Baseline              | 81.3     | 65.6     | 41.5     | 13.5     |
| DiCoNet Random Split  | 59.8     | 37.0     | 23.5     | 10.29    |
| DiCoNet               | 88.1     | 83.7     | 73.7     | 52.0     |
| DiCoNet + Split Reg   | **89.8** | **87.0** | **80.0** | **67.2** |

Table 1: ConvHull test accuracy results with the baseline PtrNet and different setups of the DiCoNet . The scale $J$ has been set to 3 for n=100 and 4 for n=200. At row 2 we observe that when the split block is not trained we get worse performance than the baseline, however, the generalization error shrinks faster on the baseline. When both blocks are trained jointly, we clearly outperform the baseline. In Row 3 the split is only trained with REINFORCE, and row 4 when we add the computational regularization term (See Supplementary) enforcing shallower trees.

## 6.2  K-means Clustering

We tackle the task of clustering a set of $n$ points with the DiCoNet in the setting described in (14). The problem consists in finding $k$ clusters of the data with respect to the Euclidean distance in $\mathbb{R}^d$. The problem reduces to solving the following combinatorial problem over input partitions $\mathcal{P}(X)$:

$$\min_{\mathcal{P}(X)} -R(\mathcal{P}(X)) = \min_{\mathcal{P}(X)} \sum_{i \in \mathcal{P}(X)} n_i \sigma_i^2 \,, \tag{15}$$

where $\sigma_i^2$ is the variance of each subset of the partition $\mathcal{P}(X)$, and $n_i$ its cardinality. We only consider the split block for this task because the combinatorial problem is over input partitions. We use a GNN (6) for the split block. The graph is created from the points in $\mathbb{R}^d$ by taking $w_{ij} = \exp\left(-\|x_i - x_j\|_2^2/\sigma^2\right)$ as weights and instantiating the embeddings with the euclidean coordinates. We test the model in two different datasets. The first one, which we call *"Gaussian"*, is constructed by sampling $k$ points in the unit square of dimension $d$, then sampling $\frac{n}{k}$ points from gaussians of variance $10^{-3}$ centered at each of the $k$ points. The second one is constructed by picking 3x3x3 random patches of the RGB images from the CIFAR-10 dataset. The baseline is a modified version of the split block in which instead of computing binary probabilities we compute a final softmax of dimensionality $k$ in order to produce a labelling over the input. We compare its performance with the DiCoNet with binary splits and $\log k$ scales where we only train with the reward of the output partition at the leaves of the tree, hence, DiCoNet is optimizing k-means (15) and not a recursive binary version of it. We show the corresponding cost ratio with Lloyd's and recursive Lloyd's (binary Lloyd's applied recursively); see Table 2. In this case, no split regularization has been added to enforce balanced partitions.

## 6.3  Knapsack

Given a set of $n$ items, each with weight $w_i \geq 0$ and value $v_i \in \mathbb{R}$, the 0-1 Knapsack problem consists in selecting the subset of the input set that maximizes the total value, so that the total weight does not exceed a given limit:

$$\begin{array}{ll} \text{maximize}_{x_i} & \sum_i x_i v_i \\ \text{subject to} & x_i \in \{0,1\}, \ \sum_i x_i w_i \leq W \,. \end{array} \tag{16}$$

| | Gaussian (d=2) | | | Gaussian (d=10) | | | CIFAR-10 patches | | |
|---|---|---|---|---|---|---|---|---|---|
| | k=4 | k=8 | k=16 | k=4 | k=8 | k=16 | k=4 | k=8 | k=16 |
| Baseline / Lloyd | **1.8** | 3.1 | 3.5 | **1.14** | **5.7** | 12.5 | **1.02** | 1.07 | 1.41 |
| DiCoNet / Lloyd | 2.3 | **2.1** | **2.1** | 1.6 | 6.3 | **8.5** | 1.04 | **1.05** | **1.2** |
| Baseline / Rec. Lloyd | **0.7** | 1.5 | 1.7 | **0.15** | **0.65** | 1.25 | **1.01** | 1.04 | 1.21 |
| DiCoNet / Rec. Lloyd | 0.9 | **1.01** | **1.02** | 0.21 | 0.72 | **0.85** | 1.02 | **1.02** | **1.07** |

Table 2: We have used $n = 20 \cdot k$ points for the *Gaussian* dataset and $n = 500$ for the CIFAR-10 patches. The baseline performs better than the DiCoNet when the number of clusters is small but DiCoNet scales better with the number of clusters. When Lloyd's performs much better than its recursive version (*"Gaussian"* with $d = 10$), we observe that DiCoNet performance is between the two. This shows that although having a recursive structure, DiCoNet is acting like a mixture of both algorithms, in other words, it is doing better than applying binary clustering at each scale. DiCoNet achieves the best results in the CIFAR-10 patches dataset, where Lloyd's and its recursive version perform similarly with respect to the k-means cost.

It is a well-known NP-hard combinatorial optimization problem, which can be solved exactly with dynamic programming using $O(nW)$ operations, referred as 'pseudo-polynomial' time in the literature. For a given approximation error $\epsilon > 0$, one can use dynamic programming to obtain a polynomial time approximation within a factor $1 - \epsilon$ of the optimal solution (Martello et al., 1999). A remarkable greedy algorithm proposed by Dantzig sorts the input elements according to the ratios $\rho_i = \frac{v_i}{w_i}$ and picks them in order until the maximum allowed weight is attained. Recently, authors considered LSTM-based models to approximate knapsack problems (Bello et al., 2016).

We instantiate our DiCoNet in this problem as follows. We use a GNN architecture as our split module, which is configured to select a subset of the input that fills a fraction $\alpha$ of the target capacity $W$. In other words, the GNN split module accepts as input a problem instance $\{(x_1, w_1), \ldots, (x_n, w_n)\}$ and outputs a probability vector $(p_1, \ldots, p_n)$. We sample from the resulting multinomial distribution without replacement until the captured total weight reaches $\alpha W$. We then fill the rest of the capacity $(1 - \alpha)W$ recursively, feeding the remaining unpicked elements to the same GNN module. We do this a number $J$ of times, and in the last call we fill all the capacity, not just the $\alpha$ fraction. The overall DiCoNet model is illustrated in Figure 3.

We generate 20000 problem instances of size $n = 50$ to train the model, and evaluate its performance on new instances of size $n = 50, 100, 200$. The weights and the values of the elements follow a uniform distribution over $[0, 1]$, and the capacities are chosen from a uniform distribution over $[0.2\,n, 0.3\,n]$. This dataset is similar to the one in (Bello et al., 2016), but has a slightly variable capacity, which we hope will help the model to generalize better. We choose $\alpha = 0.5$ in our experiments. We train the model using REINFORCE (13), and to reduce the gradient variances we consider as baseline the expected reward, approximated by the average of a group of samplings.

Table 3 reports the performance results, measured with the ratio $\frac{V_{\text{opt}}}{V_{\text{out}}}$ (so the lower the better). The baseline model is a GNN which selects the elements using a non-recursive architecture, trained using Reinforce. We verify how the non-recursive model quickly deteriorates as $n$ increases. On the other hand, the DiCoNet model performs significantly better than the considered alternatives, even for lengths $n = 100$. However, we observe that the Dantzig greedy algorithm eventually outperforms the DiCoNet for sufficiently large input $n = 200$, suggesting that further improvements may come from relaxing the scale invariance assumption, or by incorporating extra prior knowledge of the task.

This approach of the knapsack problem does not perform as good as (Bello et al., 2016) in obtaining the best approximation. However, we have presented an algorithm that relies on a rather simple 5-layer GNN, applied a fixed number of times (with quadratic complexity respect to $n$, whereas the pointer network-based LSTM model is cubic), which has proven to have other strengths, such as the ability to generalize to larger $ns$ that the one used for training. Thus, this approach illustrates well the aim of the DiCoNet. We believe that it would also be interesting for future work to combine the strengths of both aproaches.

| | n=50 | | | n=100 | | | n=200 | | |
|---|---|---|---|---|---|---|---|---|---|
| | cost | ratio | splits | cost | ratio | splits | cost | ratio | splits |
| Baseline | 19.82 | 1.0063 | 0 | 38.79 | 1.0435 | 0 | 74.71 | 1.0962 | 0 |
| DiCoNet | **19.85** | **1.0052** | 3 | **40.23** | **1.0048** | 5 | 81.09 | 1.0046 | 7 |
| Greedy | 19.73 | 1.0110 | - | 40.19 | 1.0057 | - | **81.19** | **1.0028** | - |
| *Optimum* | *19.95* | *1* | *-* | *40.42* | *1* | *-* | *81.41* | *1* | *-* |

Table 3: Performance Ratios of different models trained with $n = 50$ (and using 3 splits in the DiCoNet ) and tested for $n \in \{50, 100, 200\}$ (and different number of splits). We report the number of splits that give better performances at each $n$ for the DiCoNet . Note that for $n = 50$ the model does best with 3 splits, the same as in training, but with larger $n$ more splits give better solutions, as would be desired. Observe that even for $n = 50$, the DiCoNet architecture significatively outperforms the non-recursive model, highlighting the highly constrained nature of the problem, in which decisions over an element are highly dependent on previously chosen elements. Although the DiCoNet clearly outperforms the baseline and the Dantzig algorithm for $n \leq 100$, its performance eventually degrades at $n = 200$; see text.

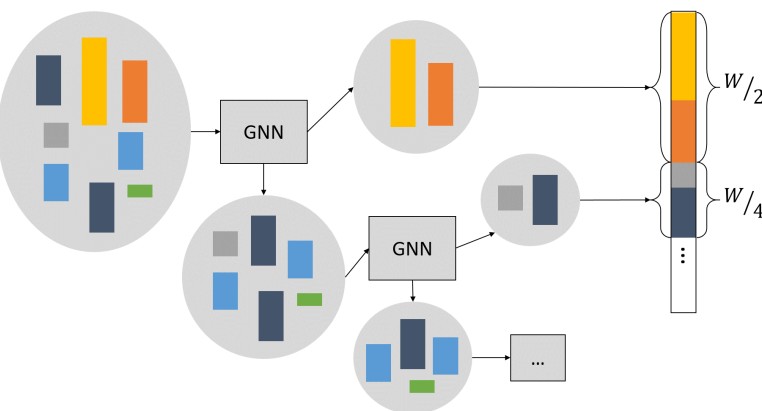

Figure 3: DiCoNet Architecture for the Knapsack problem. A GNN Split module selects a subset of input elements until a fraction $\alpha$ of the allowed budget is achieved; then the remaining elements are fed back recursively into the same Split module, until the total weight fills the allowed budget.

## 7 CONCLUSIONS

We have presented a novel neural architecture that can discover and exploit scale invariance in discrete algorithmic tasks, and can be trained with weak supervision. Our model learns how to split large inputs recursively, then learns how to solve each subproblem and finally how to merge partial solutions. The resulting parameter sharing across multiple scales yields improved generalization and sample complexity.

Due to the generality of the DiCoNet , several very different problems have been tackled, some with large and others with weak scale invariance. In all cases, our inductive bias leads to better generalization and computational complexity. An interesting perspective is to relate our scale invariance with the growing paradigm of meta-learning; that is, to what extent one could supervise the generalization across problem sizes.

In future work, we plan to extend the results of the TSP by increasing the number of splits $J$, by refining the supervised DiCoNet model with the non-differentiable TSP cost, and by exploring higher-order interactions using Graph Neural Networks defined over graph hierarchies (Lovász et al., 1989). We also plan to experiment on other NP-hard combinatorial tasks.

ACKNOWLEDGMENTS

This work was partly supported by Samsung Electronics (Improving Deep Learning using Latent Structure)

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

## A  REGULARIZATION WITH COMPUTATIONAL COMPLEXITY

As discussed previously, an added benefit of dynamic computation graphs is that one can consider computational complexity as a regularization criteria. We describe how computational complexity can be controlled in the split module.

### A.1  SPLIT REGULARIZATION

We verify from Subsection 4.1 that the cost of running each split block $\mathcal{S}$ is linear on the input size. It results that the average case complexity $C_{\mathcal{S}}(n)$ of the whole split phase on an input of size $n$ satisfies the following recursion:

$$\mathbb{E}C_{\mathcal{S}}(n) = \mathbb{E}\{C_{\mathcal{S}}(\alpha_s n) + C_{\mathcal{S}}((1 - \alpha_s)n)\} + \Theta(n) , \tag{17}$$

where $\alpha_s$ are the fraction of input elements that are respectively sent to each output. Since this fraction is input-dependent, the average case is obtained by taking expectations with respect to the underlying input distribution. Assuming without loss of generality that $\mathbb{E}(\alpha_s) \geq 0.5$, we verify that the resulting complexity is of the order of $\mathbb{E}C_{\mathcal{S}}(n) \simeq \frac{n \log n}{\log \mathbb{E}\alpha_s^{-1}}$ , which confirms the intuition that balanced partition trees ($\alpha_s = 0.5$) will lead to improved computational complexity. We can enforce $\alpha_s$ to be close to 0.5 by maximizing the variance of the split probabilities $p_\theta(z \mid X)$ computed by $\mathcal{S}_\theta$:

$$\mathcal{R}(\mathcal{S}) = - \left[ M^{-1} \sum_{m \leq M} p_\theta(z \mid X)^2 - M^{-2} \left( \sum_m p_\theta(z \mid X) \right)^2 \right] . \tag{18}$$

# B TRAINING DETAILS

The split parameters are updated with the RMSProp algorithm with initial learning rate of 0.01 and the merge parameters with Adam with initial learning rate of 0.001. Learning rates are updated as lr$/k$ where $k$ is the epoch number.

## B.1 CONVEX-HULL

Training and test datasets have 1M and 4096 examples respectively. We use a batch size of 128. The split block has 5 layers with 15 hidden units. The merge block is a GRU with 512 hidden units. The number of scales of the DCN depends on the input size. Use 0 for 6-12, 1 for 12-25 and 2 for 25-50 for training. Use 1 for 25, 2 for 50, 3 for 100 and 4 for 200 at test time. The merge block is trained using the product of the continuous $\Gamma$ matrices.

## B.2 CLUSTERING

Training and test datasets have 20K and 1K examples respectively. We use a batch size of 256. The GNN used as split block has 20 layers and feature maps of dimensionality 32.

## B.3 KNAPSACK

Training and test datasets have 20K and 1K examples respectively. The test dataset for $n = 200$ has only 100 examples, because of the difficulty to find the optimum by the pseudo-polynomic time algorithm. We use a batch size of 512. The GNN used as split block has 5 layers and feature maps of dimensionality 32.

## B.4 TSP

Training and test datasets have 20K and 1K examples respectively. We use a batch size of 32. Both the split and the merge have 20 layers with 20 hidden units. The number of scales of the DCN is fixed to 1, both for training and test time.

# C TRAVELLING SALESMAN PROBLEM

The TSP is a prime instance of a NP-hard combinatorial optimization task. Due to its important practical applications, several powerful heuristics exist in the metric TSP case, in which edges satisfy the triangular inequality. This motivates data-driven models to either generalize those heuristics to general settings, or improve them. Data-driven approaches to the TSP can be formulated in two different ways. First, one can use both the input graph and the ground truth TSP cycle to train the model to predict the ground truth. Alternatively, one can consider only the input graph and train the model to minimize the cost of the predicted cycle. The latter is more natural since it optimizes the TSP cost directly, but the cost of the predicted cycle is not differentiable w.r.t model parameters. Some authors have successfully used reinforcement learning techniques to address this issue (Dai et al., 2017), (Bello et al., 2016), although the models suffer from generalization to larger problem instances.

Here we concentrate on that generalization aspect and therefore focus on the supervised setting using ground truth cycles. We compare a baseline model that used the formulation of TSP as a Quadratic Assignment Problem to develop a Graph Neural Network (Nowak et al., 2017) with a DiCoNet model that considers split and merge modules given by separate GNNs. More precisely, the split module is a GNN that receives a graph and outputs binary probabilities for each node. This module is applied recursively until a fixed scale $J$ and the baseline is used at every final sub-graph of the partition, resulting in signals over the edges encoding possible partial solutions of the TSP. Finally, the merge module is another GNN that receives a pair of signals encoded as matrices from its leaves and returns a signal over the edges of the complete union graph.

As in Nowak et al. (2017), we generated 20k training examples and tested on 1k other instances. Each one generated by uniformly sampling $\{x_i\}_{i=1}^{20} \in [0,1]^2$. We build a complete graph with

$A_{i,j} = d_{\max} - d_2(x_i, x_j)$ as weights. The ground truth cycles are generated with Helsgaun (2006), which has an efficient implementation of the Lin-Kernighan TSP Heuristic. The architecture has 20 layers and 20 feature maps per layer, and alternates between learning node and edge features. The predicted cycles are generated with a beam search strategy with beam size of 40.

| | n=10 | | | n=20 | | | n=40 | | | n=80 | | |
|---|---|---|---|---|---|---|---|---|---|---|---|---|
| | acc | cost | ratio | acc | cost | ratio | acc | cost | ratio | acc | cost | ratio |
| BS1 | **78.71** | **2.89** | **1.01** | 27.36 | 4.74 | 1.24 | 15.06 | 9.12 | 1.77 | 13.76 | 15.07 | 2.15 |
| BS2 | 19.45 | 3.68 | 1.29 | **54.34** | **4.04** | **1.05** | 28.42 | 6.29 | 1.22 | 15.72 | 11.23 | 1.60 |
| DiCoNet | 41.82 | 3.15 | 1.11 | 43.86 | 4.06 | 1.06 | **35.46** | **6.01** | **1.18** | **29.44** | **9.01** | **1.28** |

Table 4: DiCoNet has been trained for $n = 20$ nodes and only one scale ($J = 1$). We used the pre-trained baseline for $n = 10$ as model on both leaves. BS1 and BS2 correspond to the baseline trained for $n = 10$ and $n = 20$ nodes respectively. Although for small $n$, both baselines outperform the DiCoNet , the scale invariance prior of the DiCoNet is leveraged at larger scales resulting in better results and scalability.

In Table 4 we report preliminary results of the DiCoNet for the TSP problem. Although DiCoNet and BS2 have both been trained on graphs of the same size, the dynamic model outperforms the baseline due to its powerful prior on scale invariance.

Although the scale invariance of the TSP is not as clear as in the previous problems (it is not straightforward how to use two TSP partial solutions to build a larger one), we observe that some degree of scale invariance it is enough in order to improve on scalability. As in previous experiments, the joint work of the split and merge phase is essential to construct the final solution.

## D DETAILS OF POINTER NETWORK MODULE

The merge block $\mathcal{M}_\phi$ takes as input a pair of sequences $Y_0, Y_1$ and produces an output sequence $O$. We describe first the architecture for this module, and then explain on how it is modified to perform the finest scale computation $\tilde{\mathcal{M}}_\phi$. We modify a Pointer Network (PtrNet) (Vinyals et al., 2015b) to our input-output interface as our merge block $\mathcal{M}_\phi$. A PtrNet is an auto-regressive model for tasks where the output sequence is a permutation of a subsequence of the input. The model encodes each input sequence $Y_q = (x_{1,q}, \ldots, x_{n_q,q})$, $q = 0, 1$, into a global representation $e_q := e_{q,n_q}$, $q = 0, 1$, by sequentially computing $e_{1,q}, \ldots, e_{n_q,q}$ with an RNN. Then, another RNN decodes the output sequence with initial state $d_0 = \rho(A_0 e_0 + A_1 e_1)$. The trainable parameters $\phi$ regroup to the RNN encoder and decoder parameters.

Suppose first that one has a target sequence $T = (t_1 \ldots t_S)$ for the output of the merge. In that case, we use a conditional autoregressive model of the form

$$\begin{cases} e_{q,i} = f_{\text{enc}}(e_{q,i-1}, y_{q,i}) & i = 1, \ldots, n_q , q = 0, 1 , \\ d_s = f_{\text{dec}}(d_{s-1}, t_{s-1}) & s = 1, \ldots, S \end{cases} \tag{19}$$

The conditional probability of the target given the inputs is computed by performing attention over the embeddings $e_{q,i}$ and interpreting the attention as a probability distribution over the input indexes:

$$\begin{cases} u_{q,i}^s = \phi_V^T \tanh\left(\phi_e e_{q,i} + \phi_d d_s\right) & s = 0, \ldots S , q = 0, 1 , i \leq n_q , \\ p_s = \text{softmax}(u_\cdot^s) . \end{cases} \tag{20}$$

leading to $\Gamma = (p_1, \ldots, p_S)$ . The output $O$ is expressed in terms of $\Gamma$ by binarizing its entries and multiplying it by the input:

$$, O = \mathcal{M}_\phi(Y_0, Y_1) = \bar{\Gamma} \begin{pmatrix} Y_0 \\ Y_1 \end{pmatrix} , \text{ with } \bar{\Gamma}_{s,i} = \begin{cases} 1 & \text{if } i = \arg\max_{i'} p_s(i') . \\ 0 & \text{otherwise.} \end{cases} \tag{21}$$

However, since we are interested in weakly supervised tasks, the target output only exists at the coarsest scale of the partition tree. We thus also consider a generative version $\mathcal{M}_\phi^g$ of the merge block that uses its own predictions in order to sample an output sequence. Indeed, in that case, we replace equation (19) by

$$\begin{cases} e_{q,i} = f_{\text{enc}}(e_{q,i-1}, y_{q,i}) & i = 1, \ldots, n_q , q = 0, 1 , \\ d_s = f_{\text{dec}}(d_{s-1}, y_{o_{s-1}}) & s = 1, \ldots, S \end{cases} \tag{22}$$

where $o_s$ is computed as $o_s = x_{\arg\max p_s}$, $s \leq S$. The initial merge operation at the finest scale $\tilde{\mathcal{M}}$ is defined as the previous merge module applied to the input $(X_{J,k}, \emptyset)$. We describe next how the successive merge blocks are connected so that the whole system can be evaluated and run.

## E  CONVEX HULL EXPERIMENTS

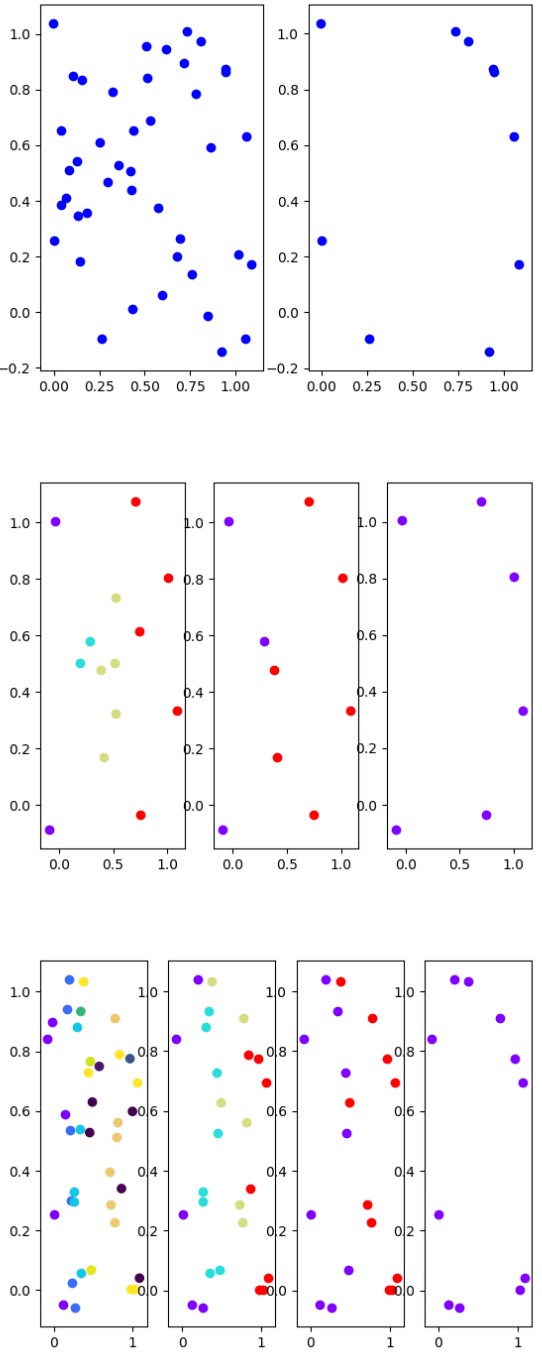

Figure 4: Output examples of the DCN at test time. Top: Baseline. Middle: DCN 1 scale. Bottom: DCN 2 scales.

