# OpenReview forum: "Divide and Conquer Networks"
_ICLR.cc/2018/Conference — Accept (Poster)_

### Official Review · AnonReviewer2 · 2017-11-25
**Neural networks enriched with divide and conquer strategy**

**Rating:** 6
**Confidence:** 3

**Review:**

This paper proposes to add new inductive bias to neural network architecture - namely a divide and conquer strategy know from algorithmics. Since introduced model has to split data into subsets, it leads to non-differentiable paths in the graph, which authors propose to tackle with RL and policy gradients. The whole model can be seen as an RL agent, trained to do splitting action on a set of instances in such a way, that jointly trained predictor T quality is maximised (and thus its current log prob: log p(Y|P(X)) becomes a reward for an RL agent). Authors claim that model like this (strengthened with pointer networks/graph nets etc. depending on the application) leads to empirical improvement on three tasks - convex hull finding, k-means clustering and on TSP.  However, while results on convex hull task are good, k-means ones use a single, artificial problem (and do not test DCN, but rather a part of it), and on TSP DCN performs significantly worse than baselines in-distribution, and is better when tested on bigger problems than it is trained on. However the generalisation scores themselves are pretty bad thus it is not clear if this can be called a success story.

I will be happy to revisit the rating if the experimental section is enriched.

Pros:
- very easy to follow idea and model
- simple merge or RL and SL in an end-to-end trainable model
- improvements over previous solutions

Cons:
- K-means experiments should not be run on artificial dataset, there are plenty of benchmarking datasets out there. In current form it is just a proof of concept experiment rather than evaluation (+ if is only for splitting, not for the entire architecture proposed). It would be also beneficial to see the score normalised by the cost found by k-means itself (say using Lloyd's method), as otherwise numbers are impossible to interpret. With normalisation, claiming that it finds 20% worse solution than k-means is indeed meaningful.
- TSP experiments show that "in distribution" DCN perform worse than baselines, and when generalising to bigger problems they fail more gracefully, however the accuracies on higher problem are pretty bad, thus it is not clear if they are significant enough to claim success. Maybe TSP is not the best application of this kind of approach (as authors state in the paper - it is not clear how merging would be applied in the first place).
- in general - experimental section should be extended, as currently the only convincing success story lies in convex hull experiments

Side notes:
- DCN is already quite commonly used abbreviation for "Deep Classifier Network" as well as "Dynamic Capacity Network", thus might be a good idea to find different name.
- please fix \cite calls to \citep, when authors name is not used as part of the sentence, for example:
Graph Neural Network Nowak et al. (2017)
should be
Graph Neural Network (Nowak et al. (2017))

# After the update

Evaluation section has been updated threefold:
- TSP experiments are now in the appendix rather than main part of the paper
- k-means experiments are Lloyd-score normalised and involve one Cifar10 clustering
- Knapsack problem has been added

Paper significantly benefited from these changes, however experimental section is still based purely on toy datasets (clustering cifar10 patches is the least toy problem, but if one claims that proposed method is a good clusterer one would have to beat actual clustering techniques to show that), and in both cases simple problem-specific baseline (Lloyd for k-means, greedy knapsack solver) beats proposed method. I can see the benefit of trainable approach here, the fact that one could in principle move towards other objectives, where deriving Lloyd alternative might be hard; however current version of the paper still does not show that.

I increased rating for the paper, however in order to put the "clear accept" mark I would expect to see at least one problem where proposed method beats all basic baselines (thus it has to either be the problem where we do not have simple algorithms for it, and then beating ML baseline is fine; or a problem where one can beat the typical heuristic approaches).

---

> ### Author Response · Authors · 2017-12-06
> **Answer to reviewer 2**
>
> First of all, we thank the reviewer for the comments.
>
> Indeed, we agree with reviewer 2 that k-means experiments should include real datasets and comparisons with Lloyd.
> We are currently working on updating the results for the k-means in order to illustrate its real performance compared to Lloyd's algorithm in a benchmarking dataset.
> We are also planning to do a small update on the TSP to multiple scales.
>
> > Side notes:
> We will consider updating the name of the paper in order to avoid conflicts with existing architectures.

---

### Official Review · AnonReviewer1 · 2017-11-27
**Good paper**

**Rating:** 7
**Confidence:** 3

**Review:**

This paper studies problems that can be solved using a dynamic programming approach and proposes a neural network architecture called Divide and Conquer Networks (DCN) to solve such problems. The network has two components: one component learns to split the problem and the other learns to combine solutions to sub-problems. Using this setup, the authors are able to beat sequence to sequence baselines on problems that are amenable to such an approach. In particular the authors test their approach on computing convex hulls, computing a minimum cost k-means clustering, and the Euclidean Traveling Salesman Problem (TSP) problem. In all three cases, the proposed solution outperforms the baselines on larger problem instances.

---

### Official Review · AnonReviewer4 · 2017-12-07
**Well written, timely idea**

**Rating:** 7
**Confidence:** 3

**Review:**

Summary of paper:

The paper proposes a unique network architecture that can learn divide-and-conquer strategies to solve algorithmic tasks.

Review:

The paper is clearly written. It is sometimes difficult to communicate ideas in this area, so I appreciate the author's effort in choosing good notation. Using an architecture to learn how to split the input, find solutions, then merge these is novel. Previous work in using recursion to solve problems (Cai 2017) used explicit supervision to learn how to split and recurse. The ideas and formalism of the merge and partition operations are valuable contributions.

The experimental side of the paper is less strong. There are good results on the convex hull problem, which is promising. There should also be a comparison to a k-means solver in the k-means section as an additional baseline. I'm also not sure TSP is an appropriate problem to demonstrate the method's effectiveness. Perhaps another problem that has an explicit divide and conquer strategy could be used instead. It would also be nice to observe failure cases of the model. This could be done by visually showing the partition constructed or seeing how the model learned to merge solutions.

This is a relatively new area to tackle, so while the experiments section could be strengthened, I think the ideas present in the paper are important and worth publishing.

Questions:

1. What is \rho on page 4? I assume it is some nonlinearity, but this was not specified.
2. On page 5, it says the merge block takes as input two sequences. I thought the merge block was defined on sets?

Typos:
1. Author's names should be enclosed in parentheses unless part of the sentence.
2. I believe "then" should be removed in the sentence "...scale invariance, then exploiting..." on page 2.

---

### Author Response · Authors · 2018-01-03
**Rebuttal**

First of all, we thank the three reviewers for their insightful comments on our work.

We have updated the paper. The main changes are:
- Changed the abbreviation from DCN to DiCoNet to avoid conflicts.
- Changed k-means split block from set2set to GNN.
- Compared k-means to Lloyd's.
- Added non-synthetic dataset for k-means: Patches of CIFAR-10 images.
- Added KnapSack problem.
- Moved TSP to appendix.

AnonReviewer4:

Comment1: There should also be a comparison to a k-means solver in the k-means section as an additional baseline.

Ans1: We agree with this comment on the k-means experimental section. We have updated the k-means
section in the following way:
      1 - We have changed the split block into a GNN to gain in expressivity (both for the DiCoNet and
           Baseline). As explained in the text, the graph is created using a Gaussian kernel.
      2 - We compare its performance with Lloyd's algorithm and Recursive Lloyd's (i.e, solving
      binary clustering recursively with Lloyd's algorithm). The performance results are shown as a ratio
      between the model costs after convergence and the algorithms output cost.
      3 - We have used a non-synthetic dataset. We have taken 3x3x3 patches of images of the CIFAR-10 dataset and applied
      the clustering models/algorithms for a pre-specified dyadic number of intervals.

Comment2: I'm also not sure TSP is an appropriate problem to demonstrate the method's effectiveness. Perhaps another problem that has an explicit divide and conquer strategy could be used instead.

Ans2: We have moved the TSP problem to the appendix and introduced the Knapsack problem, which was also
tackled in (Irwan Bello, Hieu Pham et al. '17). This problem has a clear recursive
structure, and we reaffirm this with the DiCoNet performance in the experiments.

Comment3: It would also be nice to observe failure cases of the model.

Ans3: Actually, DiCoNet performance on TSP is not that good compared to other problems due to the low
level of scale invariance compared to them.

Comment4: What is \rho on page 4? I assume it is some nonlinearity, but this was not specified.

Ans4: You are right. \rho is a pointwise non-lineariy. In particular, \rho is a sigmoid for the set2set model (split block of the convex hull), and a ReLu for the GNN.

Comment 5: On page 5, it says the merge block takes as input two sequences. I thought the merge block was defined on sets?

Ans5: The goal of the split block is to find a partition over sets (or structured sets as graphs).
The merge block takes into account the order of the previously solved instances. For instance,
in mergesort (when it merges two already ordered sequences), or the convex hull (where the previously solved instances are sequences of points ordered clockwise or counter-clockwise).

Comment6: Author's names should be enclosed in parentheses unless part of the sentence.

Ans6: Solved.

Comment7: I believe "then" should be removed in the sentence "...scale invariance, then exploiting..." on page 2.

Ans7: You are right, solved.

AnonReviewer2:

Comment8: K-means experiments should not be run on artificial dataset, there are plenty of benchmarking datasets out there. In current form it is just a proof of concept experiment rather than evaluation (+ if is only for splitting, not for the entire architecture proposed). It would be also beneficial to see the score normalised by the cost found by k-means itself (say using Lloyd's method), as otherwise numbers are impossible to interpret. With normalisation, claiming that it finds 20% worse solution than k-means is indeed meaningful.

Ans8: Same as Ans1.

Comment9: TSP experiments show that "in distribution" DCN perform worse than baselines, and when generalising to bigger problems they fail more gracefully, however the accuracies on higher problem are pretty bad, thus it is not clear if they are significant enough to claim success. Maybe TSP is not the best application of this kind of approach (as authors state in the paper - it is not clear how merging would be applied in the first place).

Ans9: We have moved the TSP section to the appendix.

Comment10: in general - experimental section should be extended, as currently the only convincing success story lies in convex hull experiments.

Ans10: We have introduced the KnapSack problem to the set of tasks and introduced an extra experiment on k-means with a non-synthetic dataset.

Comment11: DCN is already quite commonly used abbreviation for "Deep Classifier Network" as well as "Dynamic Capacity Network", thus might be a good idea to find different name.

Ans11: We have changed the abbreviation to DiCoNet.

Comment12: please fix \cite calls to \citep, when authors name is not used as part of the sentence, for example:
Graph Neural Network Nowak et al. (2017)  should be Graph Neural Network (Nowak et al. (2017))

Ans12: Solved.

---

### Decision · Program_Chairs · 2018-01-29
**ICLR 2018 Conference Acceptance Decision**

**Decision:**

Accept (Poster)

**Comment:**

The paper proposes a unique network architecture that can learn divide-and-conquer strategies to solve algorithmic tasks, mimicking a class of standard algorithms.  The paper is clearly written, and the experiments are diverse.  It also seems to point in the direction of a wider class of algorithm-inspired neural net architectures.